# Test–Retest Reliability of the Functional Electromechanical Dynamometer for Squat Exercise

**DOI:** 10.3390/ijerph20021289

**Published:** 2023-01-11

**Authors:** Indya del-Cuerpo, Daniel Jerez-Mayorga, Pedro Delgado-Floody, María Dolores Morenas-Aguilar, Luis Javier Chirosa-Ríos

**Affiliations:** 1Strength & Conditioning Laboratory, CTS-642 Research Group, Department of Physical Education and Sports, Faculty of Sports Sciences, University of Granada, 18071 Granada, Spain; 2Exercise and Rehabilitation Sciences Institute, School of Physical Therapy, Faculty of Rehabilitation Sciences, Universidad Andres Bello, Santiago 7591538, Chile; 3Department of Physical Education, Sport and Recreation, Universidad de La Frontera, Temuco 4811230, Chile

**Keywords:** reproducibility, muscle strength, dynamometer, isokinetic

## Abstract

Background: the main objective of this study was to evaluate the test-retest reliability of two different functional electromechanical dynamometry (FEMD)-controlled squat training protocols. Methods: twenty-eight healthy young adults volunteered to participate in this study. They attended the laboratory on four different days and performed four sessions: two of three sets of 12 repetitions at 75% 1RM and two of three sets of 30 repetitions at 50% 1RM. The range of movement (ROM), mean dynamic strength (MDS), peak dynamic strength (PDS), mean velocity (MV), peak velocity (PV), mean potency (MP), peak potency (PP), work (W), and impulse (I) were recorded. To evaluate the reliability of FEMD, the intraclass correlation coefficient (ICC), standard error of measurement (SEM), and coefficient of variation (CV) were obtained. Results: reliability was very high for ROM (CV: 3.72%, ICC: 0.95), MDS (CV: 1.09%, ICC: 1.00), PDS (CV: 1.97%, ICC: 1.00), and W (CV: 4.69%, ICC: 1.00) conditions at 50% 1RM and for ROM (CV: 3.90%, ICC: 0.95), MDS (CV: 0.52, ICC: 1.00), PDS (CV: 1.49%, ICC: 0.98), and W (CV: 4.14%, ICC: 1.00) conditions at 75% 1RM and high for the rest of variables at 50 and 75% 1RM. Conclusions: this study demonstrates that FEMD is a reliable instrument to measure ROM, MDS, PDS, MV, PV, PV, MP, MP, W, and I during both squat protocols (50 and 75% 1RM) in healthy young adults.

## 1. Introduction

The half squat is one of the most popular exercises for lower limb strength development in both conditioning and rehabilitation programs [1,2]. Moreover, it is not only an exercise performed during exercise routines [3] but is also frequently performed during activities of daily life, such as climbing stairs, bending down for shopping bags, or getting up from a chair [4,5,6]. To perform it in the correct way and avoid any type of injury, it is important to have an adequate range of motion and strength level [7].

Thus far, the evaluation of the parameters of strength, speed, power, work, and impulse, during the performance of this and other sporting gestures, in different research works have been evaluated independently with different devices. For example, Bosquet et al. [8] used a Musclelab device (Musclelab, Ergotest, Bjønnveien, Norway) to estimate bench press 1RM from the force-velocity relationship. Caserotti et al. [9] studied the changes in rapid muscle force, strength, and power in old and very old adults by using a force platform (Kistler, 9281 B) and an isoinertial resistance training equipment (Cybex, Medway, MA, USA). García-Ramos et al. [10] estimated the 1RM during the free-weight prone bench pull exercise through the lifts-to-failure equations proposed by Lombardi and O’Connor, general load-velocity relationships proposed by Sánchez-Medina and Loturco and the individual load-velocity relationships modeled using four (multiple-point method) or only two loads. Finally, Son et al. [11] investigated the changes in the one-legged standing balance of the ipsilateral lower limb following unilateral isokinetic strength training using an isokinetic device Biodex 3 PRO System (Biodex Medical Systems, Shirley, New York, NY, USA).

Nowadays, there is a wide range of existing multiple-joint isokinetic dynamometers (MIDs) commercially available which allow us to evaluate all these variables at once with a single device. The term MID refers to a kind of dynamometer which is specifically designed to produce a linear or quasilinear movement and record the force output of the muscles involved [12]. The MIDs are classified into “constrained” and “unconstrained,” according to the motion pattern they perform. On one hand, constrained MIDs impose a specific trajectory of motion, linear or angular. The constrained linear MIDs are subdivided into those operating as stand-alone devices, e.g., for measuring muscle performance [13,14] and as angular isokinetic dynamometry (AID) based, where AIDs are connected to specific adaptors [15,16]. Moreover, constrained angular dynamometers include stand-alone or AID-based dynamometers where the unit that provides the mechanical interface with the body moves in an angular fashion [17]. On the other hand, unconstrained MIDs provide concentric or eccentric resistance to a free, multiarticular whole-body motion, using a cable or a rod without providing proximal stabilization [18]. The cable variant incorporates a spool around which a cable is a wound, such as the functional electromechanical dynamometer (FEMD) used in this study [19,20]. The rod-based variant has a rod that operates similar to an ordinary lever arm and turns around a single ball joint, which allows for free 3D motion, though with lesser flexibility compared with its cable-based counterpart [21].

In the case of this study, to evaluate and quantify all the mentioned variables together, the FEMD has been used, a new technology that allows for evaluation and training strength together. The FEMD is characterized by its ease of use and low cost compared with the gold standard (i.e., isokinetic device), it allows it to work in dynamic (tonic, kinetic, elastic, inertial, conical) or static (isometric, vibratory) modes, allowing evaluation and training through a constant and variable resistance/velocity. Moreover, it has been shown to be a valid and reliable evaluation method [22,23]. This technology has been used to study the strength of different exercises and has obtained high-reliability values [20,23,24,25,26]. However, the reliability of the squat has not been evaluated with this device.

In fact, the reliability of this device for performing various exercises has been studied several times in recent years. It has been demonstrated that DEMF is reliable when evaluating a variety of exercises, but most of them focus on the upper body [23,24,26]. To the best of our knowledge, the exercises evaluated by DEMF involving the lower body that has been demonstrated to be reliable have lower biomechanical outputs [20,22]. Given this, we can emphasize the novelty of this study.

Therefore, the main objective of this study was to evaluate the test-retest reliability of two different FEMD-controlled squat training protocols in a group of healthy young adults. Our main hypothesis was that the protocol consisting of performing three sets of 30 repetitions of squats at 50% of 1RM has a higher test-retest reliability than three sets of 12 repetitions of squats at 75% of 1RM. This hypothesis is in line with the results obtained by Çetin et al. [27], which demonstrated that performing an exercise with loads between 60–80% 1RM resulted in a higher inter-session CV as well as a higher mean difference; and by Pérez-Castilla et al. [28] which indicated that, with loads around 85% 1RM, the CV increases as the load decreases.

## 2. Materials and Methods

### 2.1. Participants

A group of twenty-eight students of Sports Sciences (age: 25.1 ± 4.6 years; height: 1.70 ± 0.1 m; weight: 67.9 ± 13.1 kg; BMI: 23.4 ± 3.0 kg/m^2^) formed by twelve males (age: 26.1 ± 3.8 years; height: 1.78 ± 0.07 m; weight: 77.6 ± 11.3 kg; BMI: 24.6 ± 3.5 kg/m^2^) and sixteen females (age: 24.3 ± 5.1 years; height: 1.64 ± 0.07 m; weight: 77.6 ± 11.3 kg; BMI: 22.6 ± 2.54 kg/m^2^) volunteered to participate in this study. Participants were eligible for the study if (a) they had no pathology and (b) they had at least one year of experience in muscle strength training. All participants were informed of the nature, objectives, and risks associated with the experimental procedure before they gave their written consent to participate. The study protocol was approved by the Committee on Human Research of the University of Granada (no. 2182/CEIH/2021) and was conducted following the Declaration of Helsinki [29].

### 2.2. Study Design

A repeated measure design was used to determine the reliability of the squat during two different protocols (Figure 1). After the familiarization and 1RM determination session, participants attended the laboratory on four different days (at least 48 h apart) for two weeks. On each of these four days, participants performed two sessions of three sets of 12 repetitions at 75% 1RM and another two of three sets of 30 repetitions at 50% 1RM. All evaluations were performed at the same time of day (±1 h) for each participant and under similar environmental conditions (≈22 °C and ≈60% humidity). The order of the protocols was established randomly.

### 2.3. Materials

The dynamic force was evaluated with a FEMD (Dynasystem, Model Research, Granada, Spain) with an accuracy of three mm for displacement, 100 g for a detected load, and a sampling frequency of 1.000 Hz. Its control core precisely regulates both force and angular velocity using a 2000 W electric motor. The user applies forces on a rope that is wound on a roller, thus controlling, and measuring both the force and linear velocity. A load cell senses the tension applied to the rope, and the resulting signal is passed to an analog-to-digital converter with a 12-bit resolution. Displacement and velocity data are collected with a 2.500 ppr encoder attached to the roller. Data from the various sensors are obtained at a frequency of 1 kHz.

### 2.4. Familiarization Protocol and 1RM Determination

On the participants’ first visit to the laboratory, they performed a familiarization and 1RM determination session with the FEMD, and the session lasted 60 min. The familiarization consisted of (a) a general warm-up consisting of two sets of 10 squat repetitions in 2 kg increments with an initial load of 10 kg, with 40 s rest between sets, and (b) a direct estimation of the participants’ squat 1RM. For this, we started with a load of 100% of body weight in boys and 80% of body weight in girls with 4 kg increments (maximum 10 repetitions). Once this was established, there were different options: (a) the participant could perform more than one repetition, thus reaching failure. In this case, there was a rest of 5 min, and once finished, the initial load was established as the maximum load that was overcome, and increments of 1 kg were made until the resistance was insurmountable (maximum 5 repetitions). The last repetition performed was considered as the participant’s 1RM. (b) The participant could not perform any repetition. In this case, there was a rest of 2 min, and once finished, the initial load was set at 90% of body mass for boys and 70% of body weight for girls, and increments of 1 kg were made until the resistance was insurmountable (maximum 5 repetitions). The last repetition performed was considered as the participant’s 1RM. (c) The participant could only perform one repetition. There was a rest of 5 min, and once finished, the initial load was set as the same load with which it was started before, and increments of 1 kg were made until the resistance is insurmountable (maximum 5 repetitions). The last repetition performed was considered as the participant’s 1RM. (d) If the participant exceeded 120 kg (load limit of the device). We observed the number of total repetitions he/she could perform, and we estimated the 1RM with Lombardi’s equation [1] (Figure 2).

### 2.5. Evaluation Protocol

The participants arrived at the laboratory respecting the preparation conditions indicated by the investigator. They were fitted with the vest with the carabiner to which the FEMD cable was attached. After this, a warm-up was performed consisting of 5 min of cycle ergometer at an intensity of 60% of the reserve heart rate followed by 10 repetitions at 10% of 1RM to measure the angulation of the exercise. After 5 min of rest, the evaluation of three sets of 12 repetitions at 75% 1RM or 30 repetitions at 50% 1RM was performed. The order of the exercises was randomized. Between sets, a total of 5 min of rest was taken.

### 2.6. Statistical Analysis

Descriptive data are presented as mean (standard deviation) (SD). The normal distribution of the data was confirmed using the Shapiro–Wilk test (*p* > 0.05). Paired sample t-test and standardized mean differences (Cohen’s d figure effect size (ES)) were used to compare the magnitude of the load between both testing sessions. The criteria to interpret the magnitude of the ES were as follows: null (<0.20), small (0.2–0.59), moderate (0.60–1.19), large (1.20–2.00), and very large (>2.00) [30]. Test-retest reliability was assessed using the standard error of measurement (SEM) and coefficient of variation (CV), while relative reliability was assessed using the ICC, model 3.1 [30]. The following criteria were used to determine acceptable (CV ≤ 10%, ICC ≥ 0.80) and high (CV ≤ 5%, ICC ≥ 0.90) reliability [31]. Systematic bias was examined through Bland–Altman plots [32]. Finally, Pearson’s product-moment correlation coefficient (Pearson’s r) was used to quantify the correlation for all outcome variables between both testing sessions. The criteria to interpret the magnitude of the r were null (0.00–0.09), small (0.10–0.29), moderate (0.30–0.49), large (0.50–0.69), very large (0.70–0.89), nearly perfect (0.90–0.99), and perfect (1.00) [29]. For all statistical calculations, a 95% confidence interval was used in the analysis. Statistical significance was accepted at *p* < 0.05. All reliability assessments were performed by means of a customized spreadsheet [30], while other statistical analyses were performed using the JASP software (version 0.16.4).

## 3. Results

Significant differences were found during the 50% 1RM protocol, between both testing sessions, for all conditions, except for ROM (*p* = 0.068; ES = 0.00) and W (*p* = 0.429; ES = 0.26). Test-retest reliability provided stable repeatability for the range of movement (ROM), mean dynamic strength (MDS), peak dynamic strength (PDS), mean velocity (MV), peak velocity (PV), mean potency (MP), peak potency (PP), work (W), and impulse (I) condition, with a CV of less than 10% in all cases. Reliability was very high for ROM (CV: 3.72%, ICC: 0.95), MDS (CV: 1.09%, ICC: 1.00), PDS (CV: 1.97%, ICC: 1.00) and W (CV: 4.69%, ICC: 1.00) conditions at 50% 1RM (Table 1).

Furthermore, significant differences were found during the 75% 1RM protocol in the assessment of all conditions between the test and retest, except for MDS (*p* = 0.796; ES = 0.03), PDS (*p* = 0.138; ES = 0.05), and W (*p* = 0.124; ES = 0.10). Test-retest reliability provided stable repeatability for ROM, MDS, PDS, MV, PV, MP, MP, and W, with a CV of less than 10% in all cases. Reliability was very high for ROM (CV: 3.90%, ICC: 0.95), MDS (CV: 0.52, ICC: 1.00), PDS (CV: 1.49%, ICC: 0.98) and W (CV: 4.14%, ICC: 1.00) conditions at 75% 1RM (Table 2).

Bland-Altman plots for the 50% 1RM protocol reveal a low systematic bias for MDS and PDS (≤1.42 kg), MV and PV (≤7.16 cm/s), MP and PP (≤20.27 W), W (<14.32 J) and I (<40.34 kg·m/s) (Figure 3).

Bland–Altman plots for the 75% 1RM protocol reveal a low systematic bias for MDS and PDS (≤1.84 kg), MV and PV (≤8.51 cm/s), MP and PP (≤41.56 W), W (<6.62 J) and I (<83.89 kg·m/s) (Figure 4).

Finally, the r magnitude was from nearly perfect to perfect for MDS, PDS, MP, PP, and I for the 50% 1RM protocol and for all variables during the 75% 1RM protocol (r range = 0.72–0.87) (Figure 5 and Figure 6).

## 4. Discussion

The objective of this study was to evaluate the test-retest reliability of two different FEMD-controlled squat training protocols in a group of healthy young adults. The main results of this study demonstrate “near perfect” to “perfect” reliability for all variables of the 50 and 70% 1RM protocol evaluated by FEMD. These results show stable repeatability for the protocols used (CV < 10%) for all variables except for W of the 75% MR protocol, although it demonstrates a “perfect” relative reliability value (ICC = 1.00) [31].

We hypothesized that the protocol consisting of performing three sets of thirty repetitions of squats at fifty percent of one-repetition maximum (1RM) had a higher test-retest reliability than three sets of twelve repetitions of squats at seventy-five percent of 1RM. By calculating the ratio between two coefficients of variation (CVs), we were able to determine which variable from each protocol was more reliable. If the CV ratio was below 0.85 or above 1.15, it could be concluded that one protocol was more reliable than the other. Following this recommendation, we concluded that: (a) there was no difference between protocols for ROM and W; (b) the 50% 1RM protocol was more reliable for MV, PV, and I; and (c) the 75% 1RM protocol was more reliable for MDS and PDS. This means that we had to largely reject our initial hypothesis.

For many years, angular isokinetic dynamometry has been considered the gold standard for dynamic muscle performance testing [12]. With the development of new technologies, the application of multiple-joint isokinetic dynamometry is gradually increasing [32]. Within these new technologies, the DEMF would be included. Furthermore, it has been demonstrated that these devices can be validly and effectively applied for the assessment and conditioning of specific muscle activation patterns [33]. Despite this, there is a high need for standardization of testing and conditioning protocols, as well as research on the use of these training methods [12].

Nowadays, an increasing number of authors are investigating the use of these devices and the standardization of tests and training protocols [26,34]. In our case, although this is the first study that evaluates the reliability of the FEMD during the performance of a squat, the reliability of other exercises in the different working modes of the FEMD has also been evaluated in recent years. An example of this is the study by Baena-Raya et al. [22] that examined the reliability of a FEMD to assess the isometric mid-thigh pull. The results demonstrated that the PF variables calculated from the performance of the isometric mid-thigh pull on FEMD were reliable (CV < 3%; ICC > 0.90). In parallel, Reyes-Ferrada et al. [24] examined the reliability of trunk extensor strength assessment with FEMD at different velocities (0.15 m·s^−1^, 0.30 m·s^−1^, and 0.45 m·s^−1^), range of movements (25% cm and 50% cm), and isometric contraction at 90° and concluded that FEMD is a highly reliable device to evaluate trunk extensors strength. The study by Rodriguez-Perea et al. [23] determined the reliability and concurrent validity of a FEMD to measure different isokinetic velocities (0.40, 0.60, 0.80, 1.00, and 1.20 m·s^−^^1^). The results indicated that the mean velocity collected by FEMD provided a high or acceptable reliability (CV = 0.24%), as well as time to reach the isokinetic velocity (CV range = 1.68–9.70%) and time spent at the isokinetic velocity (CV range = 0.53–8.94%). Finally, Jerez-Mayorga et al. [20] determined the reliability of the strength and movement velocity of the concentric phase from the five Sit-to-Stand, using three incremental loads measured by a FEMD. The findings of this study demonstrate that FEMD is a reliable instrument to measure the average and peak strength and velocity values during the five STS in healthy young adults (ICC = 0.95–1.00; CV range: 0.79–4.18%). With all these results, the FEMD is not only reliable for the evaluation of the squat but of many other sporting gestures, types of contraction, velocities, angulations, etc.

As indicated, to evaluate a test or a training protocol, it is very important that the instrument evaluated meets certain reliability criteria [35]. It is suggested that ICC > 0.90 assures the high relative reliability of an instrument [31]. This occurs in all the variables evaluated in the two squat protocols performed in this study, so according to Weir [31], this threshold was met for all our outcome variables. These high-reliability values are due to two main reasons: (1) the high accuracy of the FEMD used, as previously stated [20,23,25,36] and (2) the familiarization process performed before the start of official data collection [20,22,23,24]. On the other hand, although there are no universally accepted thresholds for classifying CV, values below 5% are generally considered acceptable [37]. This occurs with all variables analyzed in this study for both squat protocols except for W in the 12 repetitions at 75% 1RM protocol (CV = 14.06%). This slight increase in the CV may be due to variations in the window of time during exercise performance. Despite this, Buckthorpe et al. [38] found good within-participants reliability with a slightly higher CV (CV < 19%), so it could be determined that, despite this slight increase in CV, the data for the W variable in the 12 repetitions protocol at 75% 1RM are reliable.

Dynasystem is not the only reliable FEMD. Furthermore, the reliability of other FEMD has been studied not long ago [19,36,39,40]. For example, both Cerda-Vega et al. [36] studied the validity and reliability of DEMF using three isometric strength protocols at the hip joint. The researchers reported a CV of 9.80, 6.60, and 5.64 for the side-lying, standing and supine positions, respectively. Furthermore, Campos-Jara et al. [39] demonstrated the validity and reliability while measuring the isokinetic velocity range at 0.25, 0.50, 0.75, and 1.0 m·s^−1^, using a FEMD (Haefni Health System 1.0 ^®^, Granada, Spain) compared with a linear velocity transducer (T-Force System^®^, Murcia, Spain). They reported an ICC of 0.99 for concentric and eccentric phases, while the CV was higher for the velocity of execution (1.0 m·s^−1^ = CV 4.38%). In addition, Chamorro et al. [19] investigated the test-retest reliability of FEMD at the shoulder joint. They reported an ICC value of 0.96 for 90° shoulder internal rotation and 0.94 for 90° external shoulder rotation, as well as an ICC of 0.89 for 40° shoulder internal rotation and 0.97 for 40° shoulder external rotation.

Although the high reliability of FEMD is demonstrated, this study has some limitations and considerations that should be taken into consideration for future research. Only healthy young adults whose 1RM was lower than 160 kg. Therefore, future studies should consider studying other populations, such as powerlifters, overweight or obese patients and other pathologies. Furthermore, the FEMD reliability was evaluated in half squats, and it would be interesting to know all the variable behaviors in full squats so that the participants’ 1RM would decrease and the number of participants could increase. Finally, as in most of the previous studies, the reliability of the FEMD was demonstrated. However, the validity was not investigated, and this should be addressed in the future.

## 5. Conclusions

The main findings of this study demonstrate that FEMD is a reliable instrument to measure ROM, MDS, PDS, MV, PV, PV, MP, MP, W, and I during both squat protocols (50 and 75% 1RM) in healthy young adults. In this way, with a single device and more quickly, we can reliably assess all these parameters. This allows the practitioner to have an additional alternative to record different squat evaluation variables, as well as the progress achieved through training.

## Figures and Tables

**Figure 1 ijerph-20-01289-f001:**
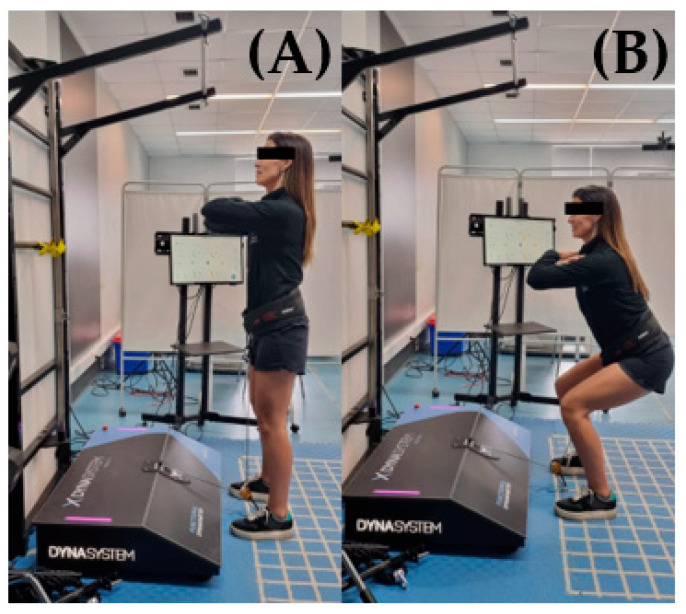
Half squat set-up. (**A**) Highest and (**B**) lowest position.

**Figure 2 ijerph-20-01289-f002:**
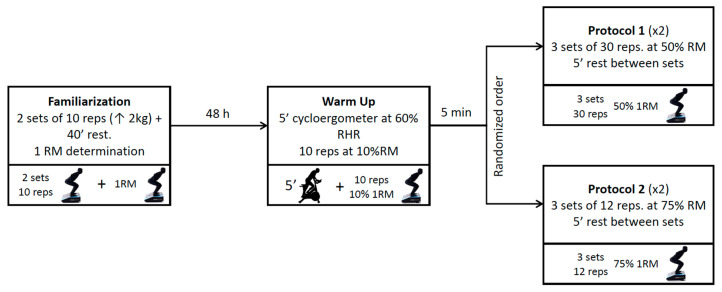
Protocol measurement of the squat exercise.

**Figure 3 ijerph-20-01289-f003:**
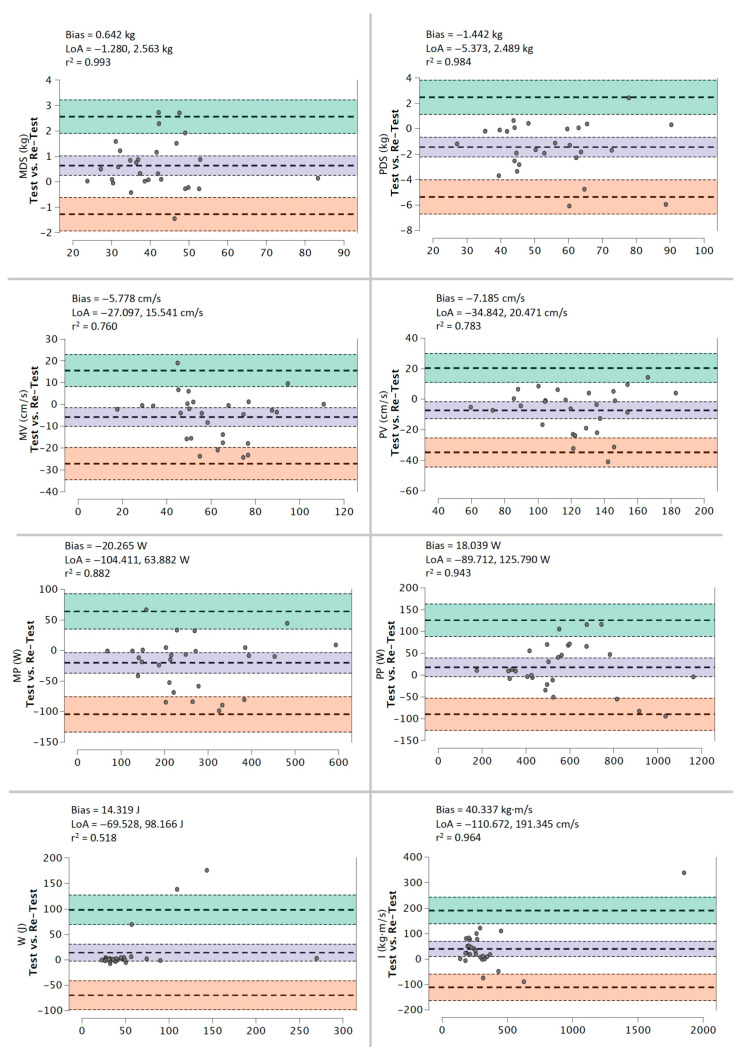
Bland-Altman plots of test–retest for MDS, PDS, MV, PV, MP, PP, W, and I during 50% 1RM protocol using a DEMF.

**Figure 4 ijerph-20-01289-f004:**
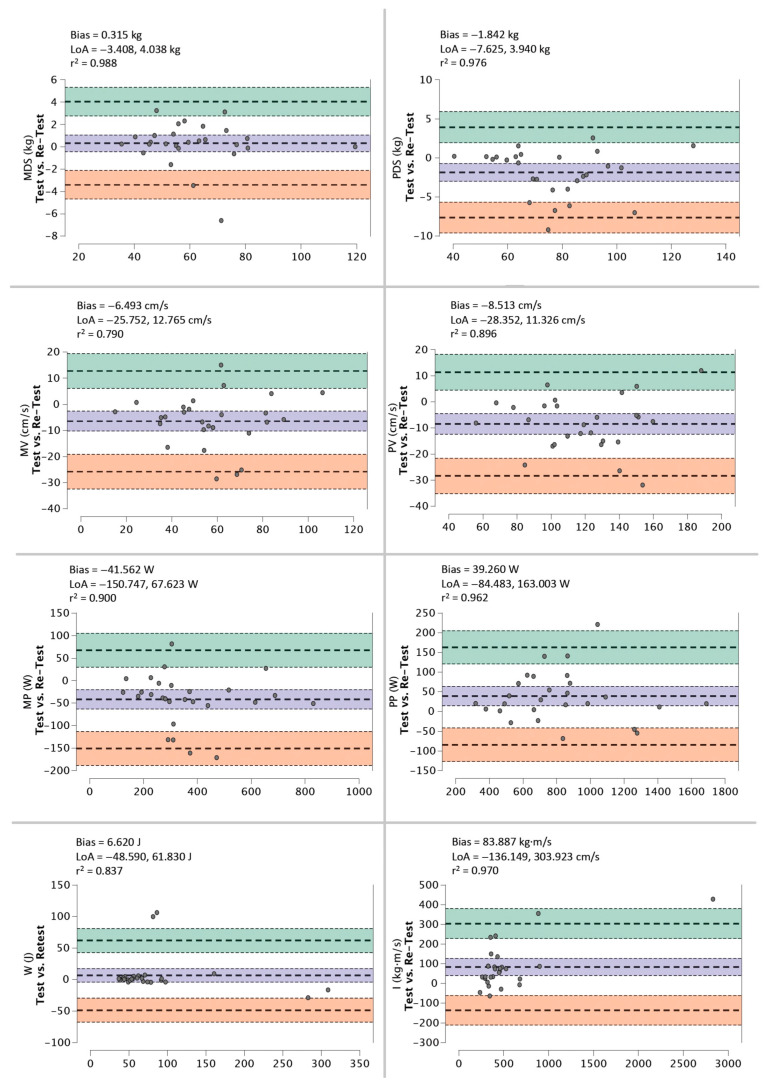
Bland-Altman plots of test–retest for MDS, PDS, MV, PV, MP, PP, W, and I during 75% 1RM protocol using a DEMF.

**Figure 5 ijerph-20-01289-f005:**
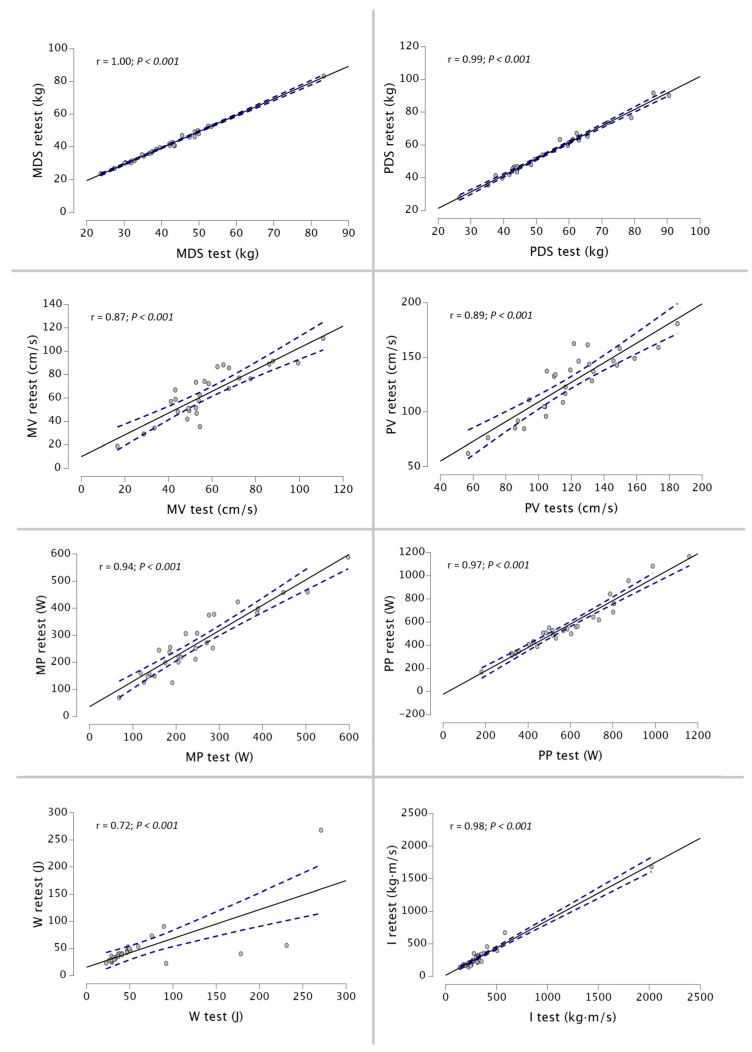
Relationship between MDS, PDS, MV, PV, MP, PP, W, and I between both testing sessions during 50% 1RM protocol using a DEMF.

**Figure 6 ijerph-20-01289-f006:**
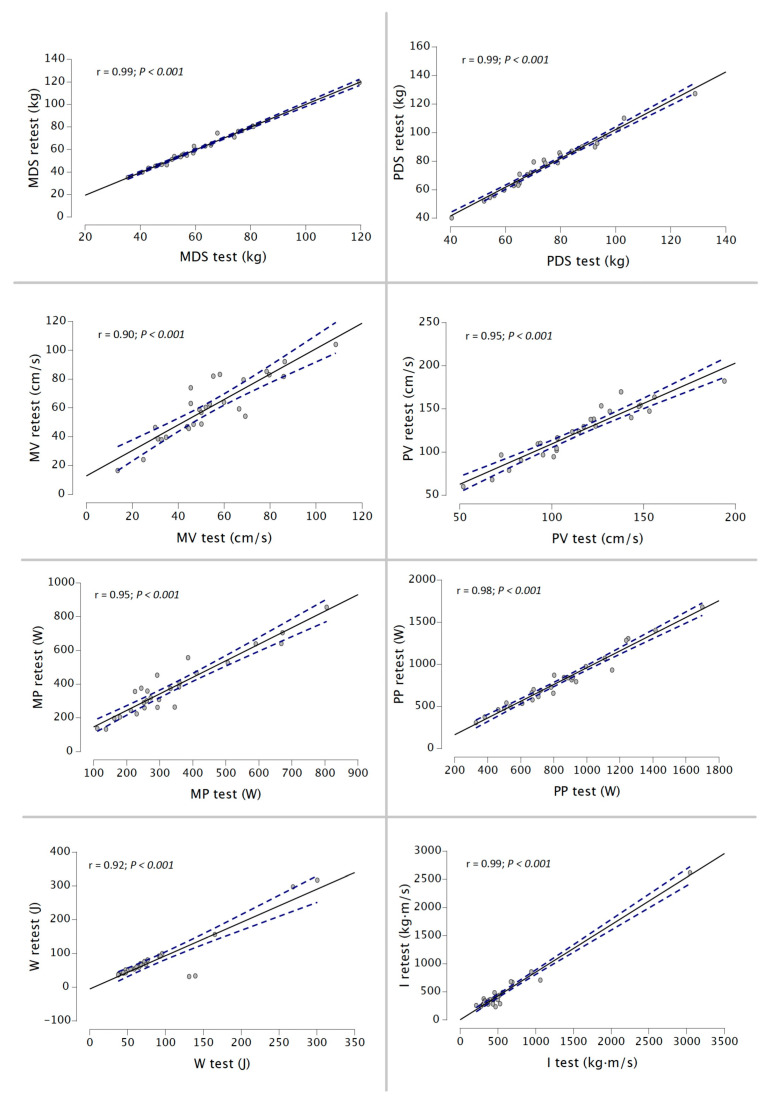
Relationship between MDS, PDS, MV, PV, MP, PP, W, and I between both testing sessions during 75% 1RM protocol using a DEMF.

**Table 1 ijerph-20-01289-t001:** Test–retest reliability of all variables during the 50% 1RM protocol using a FEMD.

		Mean ± SD		*p*-Value	ES	ICC	SEM	CV (%)
		Serie 1	Serie 2		(95% CI)	(95% CI)	(95% CI)	(95% CI)
ROM (cm)	30 reps. 50% 1RM	38.4 ± 6.0	38.4 ± 6.3	0.068	0.00 (−0.74, 0.74)	0.95 (0.89, 0.98)	0.34 (0.30, 0.39)	3.72 (3.72, 5.07)
MDS (kg)		41.4 ± 11.4	40.8 ± 11.4	0.006	0.06 (−0.80, 0.69)	1.00 (1.00, 1.00)	0.05 (0.04, 0.06)	1.09 (0.86, 1.48)
PDS (kg)		53.6 ± 15.6	55.1 ± 15.8	0.001	0.09 (−0.65, 0.83)	1.00 (0.99, 1.00)	0.08 (0.07, 0.11)	1.97 (1.46, 3.05)
MV (cm/s)		58.2 ± 21.1	64.0 ± 22.2	0.001	0.26 (−0.47, 1.01)	0.95 (0.90, 0.98)	0.23 (0.20, 0.26)	7.55 (5.97, 10.28)
PV (cm/s)		118.7 ± 29.7	125.8 ± 30.1	0.000	0.24 (−0.50, 0.98)	0.96 (0.90, 0.98)	0.26 (0.21, 0.32)	5.09 (3.76, 7.88)
MP (W)		250.2 ± 124.5	270.5 ± 123.5	0.005	0.16 (−0.58, 0.91)	0.97 (0.92, 0.99)	0.17 (0.14, 0.22)	8.36 (6.18, 12.94)
PP (W)		545.5 ± 237.5	575.7 ± 222.2	0.000	0.13 (−0.61, 0.87)	0.98 (0.95, 0.99)	0.19 (0.16, 0.24)	6.18 (4.57, 9.57)
W (J)		63.4 ± 61.6	49.1 ± 45.5	0.429	0.26 (−1.00, 0.48)	1.00 (1.00, 1.00)	0.21 (0.17, 0.26)	4.69 (3.47, 7.27)
I (kg·m/s)		353.4 ± 341.3	313.1 ± 292.7	0.005	0.13 (−0.87, 0.62)	0.99 (0.99, 1.00)	0.09 (0.08, 0.11)	8.48 (6.70, 11.54)

SD: standard deviation; ES: Cohen’s d effect size ((higher mean–lower mean)/SD both); SEM: standard error of measurement; CV: coefficient of variation; ICC: intraclass correlation coefficient; 95% CI: 95% confidence interval; ROM: range of movement; MDS: mean dynamic strength; PDS: peak dynamic strength; MV: mean velocity; PV: peak velocity; MP: mean potency; PP: peak potency; W: work and I: impulse.

**Table 2 ijerph-20-01289-t002:** Test–retest reliability of all variables during the 75% 1RM protocol using a FEMD.

		Mean ± SD		*p*-Value	ES	ICC	SEM	CV (%)
		Serie 1	Serie 2		(95% CI)	(95% CI)	(95% CI)	(95% CI)
ROM (cm)	12 reps. 75% 1RM	38.1 ± 6.5	37.7 ± 6.9	0.025	0.07 (−0.81, 0.68)	0.95 (0.90, 0.98)	0.19 (0.16, 0.22)	3.90 (3.09, 5.31)
MDS (kg)		61.0 ± 21.3	60.5 ± 21.5	0.796	0.03 (−0.76, 0.72)	1.00 (1.00, 1.00)	0.02 (0.02, 0.03)	0.52 (0.38, 0.81)
PDS (kg)		72.1 ± 21.2	73.3 ± 21.1	0.138	0.05 (−0.69, 0.79)	1.00 (0.99, 1.00)	0.07 (0.06, 0.08)	1.49 (1.09, 2.35)
MV (cm/s)		54.2 ± 21.5	60.7 ± 21.1	0.003	0.30 (−0.44, 1.05)	0.95 (0.89, 0.98)	0.25 (0.22, 0.30)	8.97 (7.09, 12.20)
PV (cm/s)		114.2 ± 32.0	122.7 ± 31.2	0.001	0.27 (−0.48, 1.01)	0.91 (0.82, 0.96)	0.24 (0.21, 0.29)	8.15 (6.44, 11.09)
MP (W)		335.1 ± 172.3	376.7 ± 176.1	0.002	0.23 (−0.50, 0.98)	0.97 (0.93, 0.99)	0.18 (0.15, 0.21)	8.84 (6.99, 12.03)
PP (W)		757.7 ± 318.3	826.5 ± 322.6	0.001	0.21 (−0.53, 0.96)	0.94 (0.88, 0.97)	0.18 (0.16, 0.22)	9.91 (7.83, 13.49)
W (J)		85.0 ± 63.8	78.3 ± 69.7	0.124	0.10 (−0.84, 0.64)	1.00 (0.99, 1.00)	0.13 (0.12, 0.16)	4.14 (3.27, 5.63)
I (kg·m/s)		564.2 ± 524.2	480.3 ± 447.1	0.067	0.17 (−0.91, 0.57)	0.98 (0.96, 0.99)	0.12 (0.11, 0.14)	14.06 (11.13, 19.16)

SD: standard deviation; ES: Cohen’s d effect size ((higher mean–lower mean)/SD both); SEM: standard error of measurement; CV: coefficient of variation; ICC: intraclass correlation coefficient; 95% CI: 95% confidence interval; ROM: range of movement; MDS: mean dynamic strength; PDS: peak dynamic strength; MV: mean velocity; PV: peak velocity; MP: mean potency; PP: peak potency; W: work and I: impulse.

## Data Availability

Not applicable.

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
