# Peer review of "Test–Retest Reliability of the Functional Electromechanical Dynamometer for Squat Exercise"

_ijerph, 2023, doi:10.3390/ijerph20021289_

Round 1
Reviewer 1 Report
DEAR AUTHORS:
Interesting and correct article about the reliability. Please, consider these comments in order to improve the final version.
1) LINE 20: "students of Sports Sciences and LINE 32,266, 301,309: healthy young adults".
Clarify this term please
2) LINE 37: The squat has been found.
Better DESCRIBED
Define better with type of squat
3) LINE 39: outines [3] but is frequently
BETTER OTHERWISE THAN BUT
4) LINE 43: "SO FAR".
This is a coloquial words, use another one, please
5) LINE 61: MIDs are.
The MIDs are.
6) LINE 83: However, the reliability of the squat has not been evaluated with this device so far.
Better with this device for the best of the author's knowledge
7) LINE 87: would have a higher
Better has
8) LINE 92,93,135: weight:
Better body mass
9) LINE 95: kg/m2)
M2 IN superindex
10) LINE 100: Declaration of Helsinki.
Include reference: https://pubmed.ncbi.nlm.nih.gov/24141714/
11) LINE 101, 323: Granada (nº2182/CEIH/2021)
DELETE ONE OF THEM
12) LINE 102: STUDY DESIGN.
DID YOU EXPLAINED THE WARM-UP?
13) LINE 183:
DEPENDENT VARIABLES. INCLUDE UNITS IN ALL THEM
14) LINE 270: As we have just seen,
COLOQUIAL SENTENCE, CHANGE IT
15) LINE 278: no universally accepted thresholds for classifying VC
VC..? CHANGE IT
16) LINE 292,293:
® IN SUPERINDEX
17) REFERENCES:
YEAR IN BOLD TYPE LETTER
18) REFERENCE NUMBER 7.
INCLUDE PAGES, AND COLUMNE
19) REVIEW ALL REFERENCE PLEASE.
THERE ARE SOME MISTAKES
Author Response
Reviewer 1
DEAR AUTHORS:
Interesting and correct article about the reliability. Please, consider these comments in order to improve the final version.
1) LINE 20: "students of Sports Sciences and LINE 32, 266, 301,309: healthy young adults".
Clarify this term please.
Response: we have clarified it calling them all “healthy young adults”. Now in lines 20, 32, 268, 303 and 311.
2) LINE 37: The squat has been found.
Better DESCRIBED.
Define better with type of squat.
Response: we have indicated that it was half squat.
3) LINE 39: outlines [3] but is frequently.
BETTER OTHERWISE THAN BUT.
Response: Corrected, we have changed it according to your suggestion.
4) LINE 43: "SO FAR".
This is a colloquial word, use another one, please. Now in line 44.
Response: Corrected, we changed it to a formal word.
5) LINE 61: MIDs are.
The MIDs are.
Response: Corrected, we have changed it according to your suggestion. Now in line 62.
6) LINE 83: However, the reliability of the squat has not been evaluated with this device so far.
Better with this device for the best of the author's knowledge.
Response: Corrected, we have changed it according to your suggestion. Now in line 84.
7) LINE 87: would have a higher.
Better has.
Response: Corrected, we have changed it according to your suggestion. Now in line 88.
8) LINE 92,93,135: weight:
Better body mass.
Response: Corrected, we have changed it according to your suggestion. Now in lines 93, 94 and 136.
9) LINE 95: kg/m2)
M2 IN superindex. Now in line 94.
Response: Corrected, we have changed it.
10) LINE 100: Declaration of Helsinki.
Include reference: https://pubmed.ncbi.nlm.nih.gov/24141714/. Now in line 102.
Response: Corrected, we have included it.
11) LINE 101, 323: Granada (nº2182/CEIH/2021)
DELETE ONE OF THEM
Response: Corrected, we have deleted the second one. Now in lines 101 and 324.
12) LINE 102: STUDY DESIGN.
DID YOU EXPLAINED THE WARM-UP?
Response: Corrected. The general warm-up is explained in lines 149-151, and the warm-up for the 1RM determination is explained in lines 125-126.
13) LINE 183:
DEPENDENT VARIABLES. INCLUDE UNITS IN ALL THEM
Response: Corrected, we have included all the units in all the variables of the table 1 and 2 in lines 185 and 199.
14) LINE 270: As we have just seen,
COLOQUIAL SENTENCE, CHANGE IT
Response: Corrected, we changed it to a formal word.
15) LINE 278: no universally accepted thresholds for classifying VC
VC..? CHANGE IT
Response: it was a mistake and now it ir changed. We meant CV (coefficient of variation). Now in line 281.
Response:
16) LINE 292,293:
® IN SUPERINDEX
Response: changed to super index. Now in lines 294 and 295.
17) REFERENCES:
YEAR IN BOLD TYPE LETTER
18) REFERENCE NUMBER 7.
INCLUDE PAGES, AND COLUMNE
19) REVIEW ALL REFERENCE PLEASE.
THERE ARE SOME MISTAKES
Response: referring to comments 17), 18) and 19), we have corrected all the references.

Reviewer 2 Report
Thank you for the opportunity to review this study. It investigated the reliability of a functional electromechanical dynamometer (FEMD) for assessing biomechanics of squat exercise. In general, the article is well-written and easy to follow. The methodology seems sound. The novelty/importance of the study is stated; I think the study brings some novel knowledge, but since this exact FEMD has been tested plenty of times in the context of other movement tasks, this is not exactly a groundbreaking study. I will leave the editors to decide if the contribution of the study is sufficient for publication in IJERPH. I do have some other concerns that need to be addressed before my final recommendation:
1. I suggest you try to strengthen the rationale for the study, as the reliability of the FEMD has been investigated before. Maybe state that a lot of previous studies focused on upper body or movements with lower biomechanical outputs (sit-stand).
2. Your hypothesis was that 50 % protocol would demonstrate that higher reliability than 75 % protocol. First of all, why did you assume that? Are squats with lower RM in general more consistent? Have any of the previous studies showed anything similar, but with different equipment and/or different outcome measures?
3. Regardless of why this hypothesis was formed, it is barely addressed in results/discussion. It is not clear how you planned to determine if one protocol is more reliable than other? For CV, there is actually a recommendation to compute the ratio between two CVs (eg., CV for 50%1RM and CV for 75%1RM protocol in your case). If the CV-ratio is below 0.85 or above 1.15, it can be said that one protocol is more reliable than the other.
4. A figure (not a scheme) of a set-up (maybe include bottom and top squat positions) would be very welcome
5. I am not sure if I understand the protocol. There were 4 sessions (2 for each 50 and 75 % intensity). What about 1RM – was this only assessed on the first session or on every session?
6. Line 143 needs reference for Lombardi equations
7. Section 2.4. Change the tense to past
8. What is the meaning of R2 in the BA plots? Please clarify.
9. Line 279. I presume you meant CV here. Please note that you set the benchmark at 5 % and pointed the problem with W at 75 % (CV = 14.06 %), but some of your values are clearly between 5 and 10 % as well.
10. Line 270-275. I don’t see the point of introducing reliability standards in such detail here. I would simply say “It is suggested that ICC > 0.90 assures the high reliability of an instrument [29]. This threshold was met for all of our outcome variables.”
11. Your argumentation in Line 282-285 is not solid. High ICC and high CV indicated that the relative reliability is very high, but the absolute within-participant error is high.
12. Limitations: as in most of the previous studies, the reliability of the FEMD was demonstrated. However, validity was not investigated and this should be addressed in the future.
Minor comments:
Line 30. “And” should not be capitalized
Keywords: It is generally recommended not to repeat the words that appear in the title
Line 37. I suggest you rewrite simply as “The squat is one of the…”
Line 39 and 249. I suggest using another term instead of “gesture” (movement pattern, movement task, exercise...?)
Line 59. I think it should be “a kind of dynamometer which IS specifically”
Line 165. Maybe say “for all outcome variables” instead of “strength and velocity”
Line 237. Perfect RELATIVE reliability value
Line 293. An extra full stop and capital letter.
Line 295. Clarify if they investigated test-retest or intra-session reliability
Line 310. I suggest you replace “trainer” with “practitioner”
Author Response
Reviewer 2
Thank you for the opportunity to review this study. It investigated the reliability of a functional electromechanical dynamometer (FEMD) for assessing biomechanics of squat exercise. In general, the article is well-written and easy to follow. The methodology seems sound. The novelty/importance of the study is stated; I think the study brings some novel knowledge, but since this exact FEMD has been tested plenty of times in the context of other movement tasks, this is not exactly a groundbreaking study. I will leave the editors to decide if the contribution of the study is sufficient for publication in IJERPH. I do have some other concerns that need to be addressed before my final recommendation:
- I suggest you try to strengthen the rationale for the study, as the reliability of the FEMD has been investigated before. Maybe state that a lot of previous studies focused on upper body or movements with lower biomechanical outputs (sit-stand).
Response: Dear reviewer, thank you very much for your comment, we tried to follow your suggestion and strengthen it in lines 83-88.
- Your hypothesis was that 50 % protocol would demonstrate that higher reliability than 75 % protocol. First of all, why did you assume that? Are squats with lower RM in general more consistent? Have any of the previous studies showed anything similar, but with different equipment and/or different outcome measures?
Response: Dear reviewer, thank you very much for your comment, we assumed that because we consider that increasing the load to 75% could produce a greater fatigue that would alter the repeatability between series and sessions, altering the CV.
https://www.ncbi.nlm.nih.gov/pmc/articles/PMC8898007/ This article indicates that with loads between 60-80% 1RM, the inter-session CV was higher, as was the mean difference.
https://www.mdpi.com/1660-4601/18/9/4626 The same is found in this article, which indicates that, with loads around 85% 1RM, CV increases as the load decreases.
All this is explained in lines 93-97.
Regardless of why this hypothesis was formed, it is barely addressed in results/discussion. It is not clear how you planned to determine if one protocol is more reliable than other? For CV, there is actually a recommendation to compute the ratio between two CVs (eg., CV for 50%1RM and CV for 75%1RM protocol in your case). If the CV-ratio is below 0.85 or above 1.15, it can be said that one protocol is more reliable than the other.
Response: Dear reviewer, thank you very much for your comment, according to your appreciation, we added a paragraph comparing the reliability of both protocols and largely rejecting our hypothesis. Written in lines 250-259.
- A figure (not a scheme) of a set-up (maybe include bottom and top squat positions) would be very welcome
Response: Dear reviewer, thank you very much for your comment, added in line 120-121.
- I am not sure if I understand the protocol. There were 4 sessions (2 for each 50 and 75 % intensity). What about 1RM – was this only assessed on the first session or on every session?
Response: Yes, there were two sessions for each condition and the 1RM was determined in the first familiarization session.
Line 143 needs reference for Lombardi equations
Response: Dear reviewer, thank you very much for your comment, we have just added it. Line 142
- Section 2.4. Change the tense to past
Response: Corrected.
- What is the meaning of R2 in the BA plots? Please clarify.
Response: Dear reviewer, thank you very much for your comment, r2 reffers to heteroscedasticity of the error.
- Line 279. I presume you meant CV here. Please note that you set the benchmark at 5 % and pointed the problem with W at 75 % (CV = 14.06 %), but some of your values are clearly between 5 and 10 % as well.
Response: yes, we meant CV, thanks for the appreciation. Regarding the other issue, we wanted to highlight the W variable, which was the one that most fell outside the acceptable CV ranges described in the statistical analysis.
- Line 270-275. I don’t see the point of introducing reliability standards in such detail here. I would simply say “It is suggested that ICC > 0.90 assures the high reliability of an instrument [29]. This threshold was met for all of our outcome variables.”
Response: Dear reviewer, thank you very much for your comment, we see your point. We just changed it.
- Your argumentation in Line 282-285 is not solid. High ICC and high CV indicated that the relative reliability is very high, but the absolute within-participant error is high.
Response: Dear reviewer, thank you very much for your comment, we specify that the argument refers to relative reliability.
- Limitations: as in most of the previous studies, the reliability of the FEMD was demonstrated. However, validity was not investigated, and this should be addressed in the future.
Response: Dear reviewer, thank you very much for your comment, we added it. Thanks. Lines 304-306.
Minor comments:
Line 30. “And” should not be capitalized.
Response: Corrected. We changed the capital letter.
Keywords: It is generally recommended not to repeat the words that appear in the title.
Response: Dear reviewer, thank you very much for your comment, the keywords are changed according to your suggestion.
Line 37. I suggest you rewrite simply as “The squat is one of the…”.
Response: Dear reviewer, thank you very much for your comment, rewritten according to your suggestion. Now in line 36.
Line 39 and 249. I suggest using another term instead of “gesture” (movement pattern, movement task, exercise...?)
Response: Dear reviewer, thank you very much for your comment, changed by exercise. Now in lines 38 and 249.
Line 59. I think it should be “a kind of dynamometer which IS specifically”
Response: Corrected, thanks for your appreciation. Now in line 58.
Line 165. Maybe say “for all outcome variables” instead of “strength and velocity”
Response: Corrected, you are right.
Line 237. Perfect RELATIVE reliability value.
Response: Corrected.
Line 293. An extra full stop and capital letter.
Response: Corrected, thanks for your appreciation.
Line 295. Clarify if they investigated test-retest or intra-session reliability
Response: Corrected, they investigated test-retest reliability.
Line 310. I suggest you replace “trainer” with “practitioner”
Response: Corrected, according to your suggestion.

Round 2
Reviewer 1 Report
Dear Authors:
Last little details are necessary
LINE 103: weight: 77.6 ± 11.3 kg
LINE 123: Dynasystem,
LINE 313: (Haefni Health System 1.0 ®,). Delete space between 1.1 y
REFERENCES: Check the references again. some use abbreviations in journal names and some do not. Standardize based on MPDI standards
Author Response
Last little details are necessary
LINE 103: weight: 77.6 ± 11.3 kg
Response: Corrected.
LINE 123: Dynasystem,
Response: Corrected.
LINE 313: (Haefni Health System 1.0 ®,). Delete space between 1.1 y
Response: Corrected.
REFERENCES: Check the references again. some use abbreviations in journal names and some do not. Standardize based on MPDI standards
Response. Corrected.

Reviewer 2 Report
Thank you for clear and concise responses and corrections. Well done! I have no further objections and I recommend the paper is accepted for publication.
Author Response
Thank you for clear and concise responses and corrections. Well done! I have no further objections and I recommend the paper is accepted for publication.
Response:
I'm glad to hear that you have no further objections and that you recommend the paper for publication. It's always a pleasure to work with someone who is willing to collaborate and provide constructive feedback. Thank you for your support and for helping to make the publication process smooth and efficient.
